# Risk of Kidney Stones: Influence of Dietary Factors, Dietary Patterns, and Vegetarian–Vegan Diets

**DOI:** 10.3390/nu12030779

**Published:** 2020-03-15

**Authors:** Pietro Manuel Ferraro, Matteo Bargagli, Alberto Trinchieri, Giovanni Gambaro

**Affiliations:** 1Università Cattolica del Sacro Cuore, 00168 Roma, Italy; pietromanuel.ferraro@unicatt.it (P.M.F.); matteo.bargagli@unicatt.it (M.B.); 2U.O.C. Nefrologia, Fondazione Policlinico Universitario A. Gemelli IRCCS, 00168 Roma, Italy; 3School of Urology, University of Milan, 20122 Milan, Italy; alberto.trinchieri@gmail.com; 4Division of Nephrology and Dialysis, Department of Medicine, University of Verona, P.le A. Stefani 1, 37126 Verona, Italy

**Keywords:** kidney stones, diet, vegetarians, vegans

## Abstract

Nephrolithiasis is a common medical condition influenced by multiple environmental factors, including diet. Since nutritional habits play a relevant role in the genesis and recurrence of kidney stones disease, dietary manipulation has become a fundamental tool for the medical management of nephrolithiasis. Dietary advice aims to reduce the majority of lithogenic risk factors, reducing the supersaturation of urine, mainly for calcium oxalate, calcium phosphate, and uric acid. For this purpose, current guidelines recommend increasing fluid intake, maintaining a balanced calcium intake, reducing dietary intake of sodium and animal proteins, and increasing intake of fruits and fibers. In this review, we analyzed the effects of each dietary factor on nephrolithiasis incidence and recurrence rate. Available scientific evidence agrees on the harmful effects of high meat/animal protein intake and low calcium diets, whereas high content of fruits and vegetables associated with a balanced intake of low-fat dairy products carries the lowest risk for incident kidney stones. Furthermore, a balanced vegetarian diet with dairy products seems to be the most protective diet for kidney stone patients. Since no study prospectively examined the effects of vegan diets on nephrolithiasis risk factors, more scientific work should be made to define the best diet for different kidney stone phenotypes.

## 1. Introduction

Nephrolithiasis is known to be a medical condition characterized by high prevalence all over the world [1,2,3]. Furthermore, during the last decades, the incidence of nephrolithiasis is rising in both genders [2,4], with resulting increased economic burden for health systems [5]. Calcium nephrolithiasis, in combination with oxalate or, less frequently, phosphate is by far the most common form, representing 75% of all kidney stone phenotypes. Conversely, the prevalence of uric acid nephrolithiasis does not exceed 10% [6]. The pathogenic pathway of calcium oxalate stone formation includes several processes (Figure 1), starting from nucleation, crystal growth, and crystal aggregation. Many factors influence urine supersaturation for calcium oxalate, being classified as promotors or inhibitors. Low urine volume, high urinary excretion of calcium, oxalate, and urate are considered as promotors. Besides, citrate, magnesium and potassium and other organic substances (nephrocalcin, urinary prothrombin fragment-1, osteopontin) are known to inhibit stone formation [7].

Many of the factors depicted in Figure 1, i.e., involved in the pathogenesis of renal stones, are influenced by the diet (Table 1). Actually, nutritional exposure is probably one of the most important factors involved in the increased frequency of nephrolithiasis among the general population. Furthermore, genetic predisposition should also be considered. More than 30 genetic variants with Mendelian inheritance are known for causing kidney stones, and polygenic involvement in idiopathic stone formers is even more frequent [8].

In addition, stone disease is associated with other comorbidities such as arterial hypertension [9], diabetes mellitus [10], obesity [11,12], metabolic syndrome [13,14], and increased likelihood of developing chronic kidney disease [15,16]. Besides, patients affected by urolithiasis have an increased odds of multi-organ complications such as metabolic bone disease [17], cardiovascular events [18,19,20], and vascular calcifications [21,22,23].

Nowadays, it is well understood that dietary advice and nutritional modifications are crucial factors in the management of nephrolithiasis and recurrence prevention [24,25,26], and, not to be forgotten, prevention of the associated systemic disorders and cardiovascular risk. In recent guidelines, dietary and medical therapies for kidney stones are commonly divided into general preventive measures and specific phenotype-based indications [25]. The first category is based on general, nonspecific indications for reducing the risk of stone formation in all kidney stone types, including increased fluid intakes, balanced calcium intake, reduced dietary intake of sodium and animal proteins, maintaining a healthy body mass index, and increasing intake of vegetables and fibers [25].

These latter indications may overlap with a vegetarian diet. Since vegetarian and vegan diets are becoming more and more popular among the general population [27], the aim of this review is to analyze each nutrient individually and to review the literature about the effect of these diets on kidney stone formation.

## 2. Methods

In this review, we included all the available literature regarding the association between vegetarian and vegan diets and kidney stone disease. We selected articles, with unrestricted search period, on several databases including PubMed, Google Scholar, the Cochrane library, and Web of Science. To identify articles of interest, we used the following search terms: “kidney stones” OR “nephrolithiasis” OR “urolithiasis” OR “kidney calculi” AND “diet”, “dietary indications”, “dietary advice”, “intake”, “vegetarian”, “vegan”. We included in this review only articles written in English language and with available full text. No restriction was made based on the type of article.

## 3. Dietary Interventions and the Risk of Stone Formation

Prevention of kidney stone recurrence is based on several medical and dietary approaches aimed at expanding urine volume, reducing the excretion of prolithogenic solutes such as calcium, oxalate, uric acid, and at increasing the excretion of substances that are protective against kidney stone formation, such as magnesium, potassium, and citrate. As a net result, the urinary supersaturation for lithogenic salts is expected to decrease. Furthermore, specific interventions are made for each kidney stone phenotype [28].

For an all-round assessment of any ongoing metabolic activity it is necessary to investigate the background causes of nephrolithiasis. High-risk patients, recurrent kidney stone formers, and also stone formers who are interested in prevention should therefore be informed of the possibility of performing 24 h urine tests for prophylactic purposes. A 24 h urine metabolic work-up is indeed capable of providing more reliable information about kidney stone risk factors than spot urine collection, having a strong correlation with kidney stone composition (Table 2) [29]. Timed urine collections are also a potentially viable alternative based on recent studies [30].

In clinical practice, the use of 24 h urine collection makes it possible to prescribe tailored dietary and lifestyle modifications based on the interpretation of urine solute excretions, avoiding unnecessary broad and difficult indications. In fact, sodium, protein, endogenous acid production, and vegetables intakes could be estimated approximately through 24 h metabolic evaluation.

Genetic work-up is available only for selected, monogenic forms of nephrolithiasis, thus for those with polygenic disposition, diet may have a variable effect and personalized dietary advice is desirable. However, since metabolic evaluation is not routinely performed in all stone formers, general dietary advice capable of reducing the risk of nephrolithiasis, avoiding bone demineralization or stunted growth in children is fundamental for kidney stones patients.

Adherence to dietary and lifestyle modification has a potentially strong impact on stone incidence, as demonstrated in a study in which body mass index (BMI), fluid intake, dietary intake of fruits and vegetables, and calcium intake explained a large proportion of first-time kidney stones [30].

Here we reviewed the effect of diet on urine solutes involved in kidney stone processes.

### 3.1. Beverages

Of all dietary interventions aimed to reduce the risk of kidney stones, fluid intake is one of the most important factors, being directly associated with the incidence of nephrolithiasis: for each 200 mL of fluids consumed per day, a 13% reduction in the risk of stone formation was found [31]. Nevertheless, since a large variety of beverages are available on the market, it is presumable that not all of them are equally protective.

Carbonated, sugar-added, soft drinks, commonly known as soda, are becoming more and more popular in the general population, especially in the USA [32]. They contain an average of 150 calories per 350 mL and are frequently fructose-added. Fructose may increase the excretion of calcium [33], oxalate [33], and uric acid [34], being associated with a higher risk of kidney stone disease [35]. For this reason, the finding of increased risk of nephrolithiasis for people with the highest consumption of soft drinks is not surprising. In detail, a study of three large cohorts (the Health Professionals Follow-Up Study, Nurses’ Health Study I, and Nurses’ Health Study II) on a total of 194,095 participants and a median follow-up of more than 8 years showed higher risk for incident kidney stones for the consumption of one or more servings of sugar-sweetened non-cola per day compared with less than one per week (+33%, 95% confidence interval (CI) = 1% to 74%), followed by sugar-sweetened cola (+23%) and punch (+18%) [36]. On the contrary, an analysis of the same cohorts suggested significantly lower risk of kidney stones for people in the highest quintile of caffeinated coffee consumption (>500 mg/day of caffeine, −26%, −29%, and −31% depending on the cohort), whereas caffeine itself seems to have an additional protective effect [37].

As regards fruit juice, grapefruit and apple juices seem to have no harmful effects on the risk of kidney stones [36]. In addition, people in the highest category of orange juice consumption have a 12% reduction in the risk of stones, compared with the lowest [36,38]. Also, a long-term intake of lemonade, consisting of 120 mL of concentrated lemon juice diluted to a 2 L solution, reduced the stone formation rate from 1.00 to 0.13 stones/patient/year [39].

Such a variability in the effect of different types of beverages could be explained by their composition: for instance, caffeine has been associated with higher urinary excretions of potassium and calcium, lower urinary oxalate excretion, and increased urine volume [37]. Lemon and orange juice have high content of citrate and are low in fructose, compared with apple juice [40,41]. Furthermore, despite lemon and orange juice having similar citrate content, only the latter seems to provide an alkali load, increasing both urinary citrate excretion, net gastrointestinal alkali absorption, and urine pH. This observation emphasizes the importance of the citrate accompanying cation: if it is a proton, as in lemonade, it could neutralize the alkalinizing effect of citrate [42].

### 3.2. Calcium

A correct calcium supplementation is essential for the osteo-articular system and muscle activity [43]. While its serum concentration is finely tuned by different systems and the kidney, its body balance is not as tightly regulated. This happens, for instance, in stone formers who frequently have a negative calcium balance due to an increase in urinary calcium excretion, which directly contributes to the genesis of nephrolithiasis [44].

It could be presumed that reducing dietary calcium intake is an appropriate therapeutic strategy (Table 3). However, it should be considered that calcium intake rarely exceeds 1.2 g/day with diet [45].

Significant insights on the role of calcium in stone disease were obtained in 1993, when an analysis of 45,619 40–75 years old male individuals without stone disease at recruitment (the Health Professionals Follow-up Study cohort) showed that lower calcium intakes were associated with higher risk of kidney stone events by more than 50% compared with higher dietary intakes (797 ± 280 vs. 851 ± 307 mg of dietary calcium, respectively) [46]. This was confirmed in a randomized trial comparing a normal calcium (1200 mg/day), low-salt, and low-animal-protein diet to a low-calcium diet (400 mg/day) in a group of 120 men with recurrent calcium oxalate stones and hypercalciuria. In this study, the normal calcium diet group demonstrated a significant reduction in the risk of stone recurrence of approximately 50% after five years compared with the low-calcium diet group (relative risk for recurrence 0.49, 95% CI 0.24 to 0.98). In addition, urinary oxalate excretion was higher in patients on low dietary calcium intake (60 µmol/day increase) and lower in patients under normal calcium, low-salt, and low-animal-protein diet (80 µmol/day decrease) [47]. This phenomenon is explained by the fact that calcium in the intestine acts as a chelator for several substances, including oxalate. In case of low-calcium diet, there is an increased intestinal absorption of free oxalate, which increases oxaluria and the urinary supersaturation for calcium oxalate, favoring the nucleation process [48]. Furthermore, balanced dietary calcium intake seems to have protective effects on kidney stone events independent of its origin, from both dairy and nondairy sources [49].

### 3.3. Oxalate

Oxalate is mainly found in plants, which use it to eliminate excessive calcium present in water. In fact, it accumulates in leaves, fruits, and seeds. When those parts are detached, the calcium excess is eliminated together with oxalate [50].

For this reason, large quantities of oxalate are usually ingested every day, although the exact amount is difficult to estimate. A particular example of high variability in oxalate content of foods is tea: black tea has higher oxalate concentration compared with oolong or green tea. Besides, other factors such as brewing time, tea quality, preparation, origin, and harvesting period influence urinary oxalate excretion [51,52,53].

In any case, only 50% of normal daily urinary oxalate excretion is of food origin; the remaining amount is due to endogenous hepatic metabolism [54]. This molecule has no nutritional function and is therefore eliminated through the kidneys. In urine, it avidly binds calcium, increasing calcium oxalate supersaturation [54,55].

Normally, intestinal oxalate absorption is low and highly variable (around 10%–15%) [56]. In individuals without malabsorption syndrome, bowel oxalate uptake may increase only when intestinal ionized Ca is reduced, often due to high dietary consumption of phytate (calcium-binding molecule) and/or low-calcium diet [57].

High oxalate foods include broadleaf vegetables such as spinach, green cabbage, beets, but also nuts, tea, chocolate, and rhubarb. Oxalate is anyway broadly present in foods, so it is difficult to significantly limit its intake. However, as a general indication to administer to stone patients, dietary manipulation might be useful only in case of excessive intake of high-oxalate vegetables, such as spinach, chocolate, and nuts (Table 4).

The type of diet and the oxalate content of foods are equally important in determining urinary oxalate excretion: Borghi et al. proved that an increased intake of vegetables in a mixed diet provides increased urine citrate excretion without any negative effects on oxalate excretion as well [58].

Although clinical trials are underway, there are currently no available agents on the market capable to decrease intestinal oxalate.

The use of probiotics such as VSL-3 (mixture of eight categories of lactic acid-producing bacteria) to reduce intestinal oxalate is controversial [59]. Since gut microbiome may affect nephrolithiasis by modulating the quantity of oxalate absorbed from the gut and afterward excreted in urine [60], the reintegration of a physiological intestinal bacterial flora could increase the intestinal degradation of oxalate. However, although differences in gut microbiome composition between stone formers vs. non-stone formers were recently found [61], there is still lack of solid scientific evidence for the effectiveness of probiotics in reducing the urinary excretion of oxalate. Much of the attention is focused to the use of enzymes derived from *Oxalobacter formigenes*, a microorganism capable of degrading intestinal oxalate [62]. However, this therapeutic approach is still under investigation.

Thus, the only way now available to reduce oxalate intestinal uptake is to use calcium supplements or dietary sources of calcium, in case of meals high in oxalate [63].

### 3.4. Sodium

High dietary intake of sodium chloride is a cause of worsening blood pressure control and increased heart disease but is also directly associated with urinary calcium excretion. Patients affected by idiopathic hypercalciuria have indeed higher salt intake than other stone formers [64]. Even for healthy subjects, the higher the sodium load, the greater the calcium excretion. In fact, a 6 g increase in daily dietary sodium chloride seems to be capable of increasing urinary calcium excretion by 40 mg/day [65,66]. Furthermore, a dietary salt intake >10 g/day was correlated to increased prevalence of hypercalciuria compared with recommended values [67].

Although the daily intake of sodium necessary for the body homeostasis is around 0.5 g (21.8 mmol) [68], it is normally ingested in higher doses. The average daily intake in an Italian population was 10.9 g/day (186 mmol/day) in men and 8.5 g/day (145 mmol/day) in women [69], whereas a single teaspoon of table salt contains about 2.3 g (100 mmol) of sodium, representing the maximum recommended daily dose [70].

Several studies evaluated the effect of dietary sodium restriction on the risk of stone formation. Lin et al., in a randomized controlled trial, showed that reducing sodium intake from 150 to 50 mmol/day is correlated to a 0.5 mmol/day decrease in urinary calcium excretion [71]. This evidence is supported by a more recent study, demonstrating the effectiveness of the DASH diet (Dietary Approaches to Stop Hypertension, low sodium intake and high fruits and vegetables assumption) in reducing the incidence of kidney stone events [72]. In conclusion, in case of high urinary calcium excretion with a high estimated dietary sodium (urinary sodium excretion >200 mmol/day) it is always advisable to introduce a low-sodium diet. However, this can be very complicated to achieve with processed foods as this element is added to virtually any industrial-transformed food (Table 5).

### 3.5. Proteins of Animal and Plant Origin

Proteins are essential for correct nutrition and their unique properties influence also kidney stone disease. High dietary content of non-dairy animal proteins (poultry, meat, fish, eggs) together with low-alkali food are thought to be deleterious to kidney stone formers, causing a negative calcium balance, low urinary pH, and low urinary excretions of citrate, potassium, and magnesium [71,73,74,75]. Animal proteins increase purine metabolism, contributing to hyperuricosuria in both uric acid and calcium nephrolithiasis [76]. In addition, high animal protein intake seems to affect also urinary oxalate excretion in the setting of idiopathic calcium nephrolithiasis with or without mild metabolic hyperoxaluria [77]. On the other hand, no significant interaction was found between meat and urine oxalate in healthy subjects [77].

Epidemiological evidence supports the association between protein intake and kidney stone risk: we previously [30] showed that a DASH-style diet, rich in vegetables and low in animal proteins, has the lowest risk of incident kidney stones in three large prospective cohorts. In fact, it was estimated that consuming high amounts of fruits and vegetables in addition to low-fat dairy products is capable of lowering the risk of stone events by up to a 45% [72].

Dietary protein restriction alone can also have a favorable effect on metabolic risk factors for nephrolithiasis. Indeed, Giannini et al. demonstrated that in 18 hypercalciuric stone formers a protein intake of 0.8 g/kg/day and 955 mg of calcium for 15 days significantly improved several urinary lithogenic risk factors, i.e., decreased urinary calcium (from 9.35 ± 0.3 to 6.45 ± 0.3 mmol/day), oxalate (from 0.59 ± 0.09 to 0.31 ± 0.03 mmol/day), and uric acid excretion (from 3.1 ± 0.1 to 2.5 ± 0.1 mmol/day) and increased urinary citrate excretion (from 3.42 ± 0.3 to 5.34 ± 0.9 mmol/day) [78].

The protein source should also be taken into account. In a study on three large cohorts, vegetable proteins were not associated with the risk of kidney stones, even after adjustment for age, BMI, or the use of thiazides, and the higher potassium intake due to high vegetable intake was one of the most protective factors for nephrolithiasis [79]. At the same time, the intake of dairy protein was inversely associated with incident kidney stone disease. Only nondairy animal proteins seem to be harmful for the occurrence of kidney stones with a hazard ratio ranging from 1.15 to 1.20 in different cohorts [79].

### 3.6. Citrate, Dietary Alkali Load and Magnesium

Intake of fruits and vegetables is fundamental in kidney stone formers to provide a sufficient amount of dietary alkali and citrate supplementation. The beneficial effects of fruits and vegetables on kidney stone risk is linked also to their alkalizing abilities. In fact, metabolic acidosis is known to upregulate tubular reabsorption and metabolization of citrate through the sodium-dependent dicarboxylate transporter 1 (NaDC-1) and the cytosolic ATP citrate lyase, whereas alkalosis or citrate administration downregulates these enzymes, thus increasing urine citrate excretion [80]. According to these evidences, the amount of acids excreted by the kidneys every day is connected to the risk of stones. Using a validated formula, it is possible to estimate the net renal acid excretion from diet, and a food screener for the rapid determination of the potential renal acid load (PRAL) has been developed [81,82]. Higher PRAL of food is directly associated with calciuria [83], with citraturia [84], and ultimately with increased risk of stone formation [85,86].

Citrate has an important role in urinary alkalization and antilithogenic activity. It has a high affinity for calcium, inhibiting crystallization of calcium crystals. Citrate also prevents the aggregation of already formed calcium oxalate crystals, thus preventing the formation of bigger concretions and stones.

Due to their elevated potassium citrate concentrations, it was demonstrated that drinking 1.2 L of orange juice or 2 L of lemon juice per day increases urinary citrate excretion and reduces kidney stones recurrence rate in both normal subjects and hypocitraturic stone formers [36,40,41,87].

We previously demonstrated that dietary potassium also has a strong relationship with the incidence of nephrolithiasis [79]. However, the origin of the potassium intake should also be considered: only potassium citrate, but not potassium chloride, was capable of reducing urinary calcium excretion in healthy subjects [88]. Furthermore, we showed that only increased dietary animal protein to potassium ratio was associated with a greater risk of incident nephrolithiasis, whereas vegetable proteins have no significant association to stone risk [79]. In fact, meat is also a source of dietary potassium, but its acidifying effect due to high content of sulfuric amino acids limits its antilithogenic effect.

These observations led to the modern dietary indications for calcium stone formers: a diet rich in fruits and vegetables, with low animal protein and salt intake. Meschi et al. [58], in 2004, analyzed the effect of dietary alkali load on urinary composition in both healthy individuals and stone formers. They proved that withdrawing fruits and vegetables from the diet for 14 days in 12 normal subjects significantly reduced urinary excretion of magnesium (−26%), citrate (−44%), potassium (−62%), and oxalate (−31%) and increased urinary calcium (+49%) and ammonium (+12%). Urine supersaturation for calcium oxalate and phosphate increased as well (from 6.33 to 8.24 and from 0.68 to 1.58, respectively). On the contrary, a diet rich in fruits and vegetables in hypocitraturic stone formers increased urinary pH (from 5.84 to 6.19), excretion of potassium (+68%), citrate (+68%), and magnesium (+23%) and reduced ammonium (−18%), whereas urinary excretion of calcium and oxalate remained unchanged. These urinary modifications led to reduced supersaturation for uric acid and calcium oxalate (from 10.17 to 4.96 and from 2.78 to 1.12, respectively) [58].

Fruits and vegetables are also the main source of magnesium. The antilithogenic effect of magnesium has been proven in vitro: it can inhibit calcium oxalate crystal formations in urine, binding free oxalate and increasing its solubility [89]. Furthermore, a recent study demonstrated that magnesium decreases both calcium oxalate and calcium phosphate aggregation in a concentration-dependent manner. These activities are amplified by citrate and work even at acidic pH [90]. Magnesium works also as a chelator of oxalate in the bowel, thus reducing its intestinal absorption. Despite the strong rationale for considering magnesium a kidney stone inhibitor, the available literature is controversial. Early observational evidence showed no significant association between magnesium and the risk of stones [46,67], whereas Taylor et al., in 2004, demonstrated an inverse correlation between the risk of incident kidney stones and dietary magnesium intake [91].

### 3.7. Uric Acid

In recent years, metabolic syndrome has been found to be the predominant cause of uric acid nephrolithiasis [13,92]. The underlying pathophysiological process has resulted from unduly acidic urine pH, linked to insulin resistance [93] and to a lesser extent from a high dietary purine content and increased endogenous purine metabolism with consequent overproduction of uric acid [94].

Whatever the cause, meals with a high animal meat content (purine load) tend to increase the filtered uric acid load. For this reason, it is advisable to reduce the intake of beef, pork, shellfish, fish, and chicken. Red and white meat are equivalent in purine load.

Not only an increase in uricosuria but also a low urinary pH predisposes to the precipitation of uric acid crystals. As already mentioned, high animal protein intake is a purine source, but it also has acidifying properties. In fact, Siener et al. showed that a vegetarian diet composed of low animal proteins and high fluid, fruits, and vegetables intakes has the lower risk of uric acid crystals formation compared with an omnivorous diet [95].

## 4. Impact of Vegetarian/Vegan Diets on Kidney Stone Recurrence and Prevention

The definition of a vegetarian diet, according to the Vegetarian Society, is a diet without any form of animal slaughter products such as meat, poultry, fish, shellfish, and game [96]. The majority of vegetarian diets allow the consumption of dairy products and eggs in addition to high levels of fruits, vegetables, grains, nuts, and seeds (lacto-ovo-vegetarianism). However, the ovo-vegetarianism variant excludes also dairy products [97]. On the contrary, the vegan diet is a very strict form of vegetarian diet that does not allow the use of any form of animal-derived products. After this brief review of the main dietary factors influencing kidney stone formation and prevention, we can conclude that available scientific evidence highlights that a diet high in fruits and vegetables, low in animal proteins, balanced in low-fat dairy products, and with a reduced salt content is the best way to decrease the risk of kidney stone disease (Table 4). These characteristics match the definition of a balanced vegetarian diet, allowing consumption of dairy products. On the contrary, a vegan diet, which is rich in oxalate from plants and is poor in calcium because dairy products are not allowed, should be less protective and, at least in theory, might favor calcium oxalate lithogenesis.

Accordingly, in the vegetarian diet, calcium intake seems to be comparable to controls, whereas in vegan diet the dietary consumption of calcium is lower [98,99].

Currently, no studies are available that directly compare the effect of vegan and vegetarian diets on incident kidney stones. A possible beneficial effect of vegetarianism was first suggested by Robertson, predicting a 40%–60% reduction of kidney stone prevalence compared with the general population, matched for age, sex, and social class [100,101].

An analysis of the Oxford cohort of the European Prospective Investigation into Cancer and Nutrition (EPIC) compared the association between five different types of diets and kidney stones development: high (>100 g/day), moderate (99–50 g/day), and low (<50 g/day) meat intake, fish-eaters, and vegetarian diets [102]. This study was conducted on 51,336 participants (23% were male) aged ≥20 years. A total of 303 episodes of incident kidney stones were recorded (716,105 person-years of follow-up). It turned out that, compared with high-meat-intake diet, vegetarian and low-meat-intake diets had the lowest hazard ratio: 0.69 (95% CI 0.48, 0.98) and 0.52 (95% CI 0.35, 0.80), respectively. Furthermore, both red meat and poultry were significantly correlated to increased risk of stone events, whereas processed meat was not. As expected, high fruits intake showed an inverse association with the risk of stone formation.

These observations were confirmed by studies investigating the effect of the DASH-style and the Mediterranean diets on renal stones. In one large prospective study of 241,766 male and female health workers in the USA, a score for the DASH-style diet was devised based on eight components (intake of fruits, vegetables, nuts and legumes, low-fat dairy products, whole grains, intake of sodium, sweetened beverages, and red meats). This was used to assess the diet over 14–18 years. Comparing individuals in the highest and lowest DASH score quintiles, those with the highest scores had significantly lower risk for developing kidney stones. Multivariate relative risks for stones were 0.55, 0.58, and 0.60 in men, older and younger women, respectively, compared with regular American diet [72].

The Mediterranean diet, which is characterized by low meat intake and is plant-based, brings the same protective properties. In fact, in a study on a cohort of 16,094 subjects without history of kidney stones, the adherence to the Mediterranean diet was evaluated through a food frequency questionnaire, and the hazard ratios of the two groups with the highest adherence to the Mediterranean diet were 0.93 (95% CI 0.79, 1.09) and 0.64 (95% CI 0.48, 0.87), compared with the less adherent group [103]. Interestingly, in this cohort, patients with higher monounsaturated fatty acid to saturated fatty acid ratio had a greater risk of stone events compared with patients with higher intake of vegetables and dairy products [103]. This is a further advantage of the Mediterranean diet because of the higher cardiovascular and end-stage kidney risks of stone patients. Actually, the Mediterranean diet has been shown to decrease both [104,105].

A case-control study of 1019 kidney stone formers and 987 healthy controls in China suggested that several “vegetarian-oriented” foods other than meat may increase the risk of nephrolithiasis. Grains consumption and beans products were positively associated with kidney stones (odds ratio [OR] 2.08; 95% CI 1.08, 4.02 and OR 3.50; 95% CI 1.61, 7.59, respectively) in women. Furthermore, elevated intake of leafy vegetables (more than 3 servings/day) was directly correlated with stones in both genders (OR for men 2.02; 95% CI 1.04, 3.91; for women 3.86; 95% CI 1.48, 10.04) [106].

The effect of fibers on kidney stone disease is controversial. Gleeson et al., in 1990 [107], showed that a fiber-rich diet, if associated with low calcium intake, lowered urinary calcium excretion at the expense of increased urinary oxalate concentration, with uncertain effects on calcium oxalate supersaturation [106]. More recently, Littlejohns et al. [31] showed that fibers intake was protective for kidney stone disease, although vegetables intake was neither protective nor harmful. These results were confirmed in the EPIC-Oxford analysis [102].

## 5. Conclusions

Diet is considered a fundamental tool for the management of kidney stone recurrence [102]. Above all dietary indications and the possibility to give patient-tailored dietary suggestions [108], maintaining an elevated fluid intake [25] is the most important element for kidney stone prevention.

In this review, we analyzed the effects of each dietary factor on nephrolithiasis incidence and recurrence rate. Current scientific evidence agrees on the harmful effects of high meat/animal protein intake and low-calcium diets, whereas high content of fruits and vegetables associated with a balanced assumption of low-fat dairy products carries the lowest risk for incident kidney stones. Based on the available evidence, a balanced vegetarian diet with dairy products seems to be the most protective diet for kidney stone patients. Besides, the effect of vegetables and fibers alone on the risk of stone formation is not yet completely known. It is important to keep in mind that teasing out the effect of a single nutrient is rather difficult because they interact with each other, modifying the risk of kidney stones. Since no study prospectively examined the potential protective effects of vegan diets on nephrolithiasis risk factors, more scientific work should be made to define the best diet for different kidney stone phenotypes.

## Figures and Tables

**Figure 1 nutrients-12-00779-f001:**
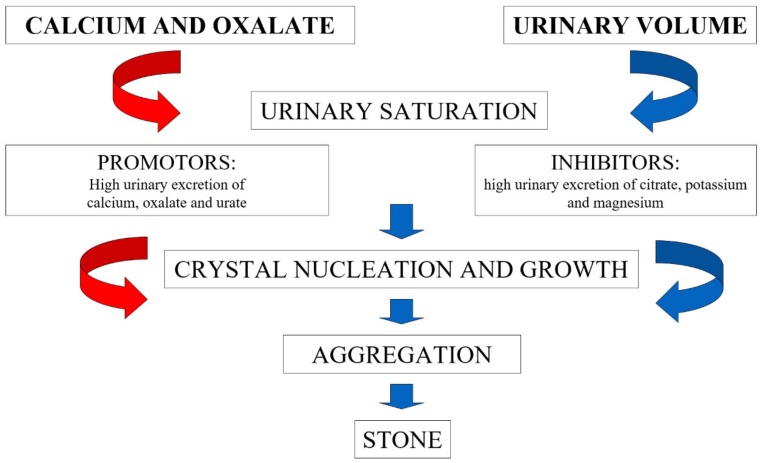
Mechanisms of calcium oxalate stone formation.

**Table 1 nutrients-12-00779-t001:** Dietary factors and potential stone risk.

Dietary Factors	Modification	Potential Stone Risk
Fluid intake	Reduction	Increased urine saturation
Sodium intake	Increase	Increased urine calcium and reduced citrate excretion
Calcium intake	Reduction	Increased urinary oxalate excretion
Meat intake	Increase	Low urine pH, increased urine calcium and reduced citrate excretion
Fruits intake	Reduction	Low urine pH and reduced citrate excretion
Diet content in oxalate foods	Increase	Increased urinary oxalate excretion

**Table 2 nutrients-12-00779-t002:** Dietary advice for single 24 h urine abnormalities.

Hypercalciuria (>6.5 and 7.5 mmol/day for women and men, respectively)
Low animal protein intake (0.8–1 g/kg/day)
Low salt intake (<5 g NaCl/day)
High intake of fruits and vegetables
**Hyperoxaluria (>0.5 mmol/day)**
Low dietary oxalate intake
Balanced calcium intake (1.2 g/day)/calcium supplement
**Hypocitraturia** **(< 1.5 mmol/day)**
Low animal protein intake (0.8–1 g/kg/day)
High intake of fruits and vegetables/potassium, citrate, and magnesium supplementation
**Low urine pH/hyperuricosuria (>4.5 and 4.8 mmol/day for women and men, respectively)**
High intake of fruits and vegetables/potassium, citrate, and magnesium supplementation
Low dietary purine intake
Low animal protein intake (0.8–1 g/kg/day)

**Table 3 nutrients-12-00779-t003:** Calcium content of foods.

Food Category	Calcium Content/Serving (mg)
**Dairy Products**	
Yogurt	415
Milk	276
Cheese	138–333
**Fish**	
Sardines	286
Salmon	179–212
**Sulfate-containing vegetables**	
Tofu	138
Turnip greens	99
Kale	94
Bok choi	74
Broccoli	21
**Vegetables rich in oxalate**	
Soybeans	175
Spinach	154
White beans	93–141
Almonds	93

**Table 4 nutrients-12-00779-t004:** List of foods rich in oxalate.

Foods at very high content of oxalate (>200 mg/serving)
Rhubarb
Spinach
Rice bran
**Foods at high content of oxalate (>50 mg/serving)**
Beets
Hibiscus esculentus (Okra, Gombo, Bamia)
Potatoes (with skin)
Beans
Almonds
Cocoa powder
Wholegrain cereals
Miso soup
Wheat bran
Cornmeal
Oat bran
Pseudocereals (quinoa, amaranth)

**Table 5 nutrients-12-00779-t005:** Foods with the highest content of sodium.

High Sodium Foods
Smoked, cured, or canned meat (bacon, ham, sausages)
Smoked, cured, or canned fish (sardines, caviar, anchovies)
Salted beans
Olives
Prepackaged or canned entrees (ravioli, spam, chili)
Cheese (roquefort, parmesan)
Salted nuts
Soy sauce
Mustard
Ketchup
Bread with salted tops
Croutons
Saltine crackers

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
