# Peer review of "Risk of Kidney Stones: Influence of Dietary Factors, Dietary Patterns, and Vegetarian–Vegan Diets"

_nutrients, 2020, doi:10.3390/nu12030779_

Round 1
Reviewer 1 Report
This is the first submission of a Review paper entitled "Risk of kidney Stones: Influence of Dietary Factors, Dietary Patterns and Vegetarians-Vegans Diets". The main purpose of this work was to review the available literature on the effect of diet on kidney stone formation.
The paper is well written and gives a good summary of the topic.
The authors should consider adding a section describing how they use information on the various stone phenotypes and results of urine metabolic risk factor evaluation (how they interpret 24 hour urine sample results) in their clinical practice to guide their dietary recommendations. An annotated Figure describing their current clinical algorithm?
Author Response
The authors should consider adding a section describing how they use information on the various stone phenotypes and results of urine metabolic risk factor evaluation (how they interpret 24 hour urine sample results) in their clinical practice to guide their dietary recommendations. An annotated Figure describing their current clinical algorithm?
Answer: We added a table, analysing dietary indications for single 24h urine abnormalities:
Table 2. Dietary advice for single 24h urine abnormalities
|
Hypercalciuria (> 6.5 and 7.5 mmol/day for women and men respectively) |
|
Low animal protein intake (0.8 – 1 g/Kg/day) |
|
Low salt intake (< 5 g NaCl / day) |
|
High intake of fruits and vegetables |
|
Hyperoxaluria (> 0.5 mmol/day) |
|
Low dietary oxalate intake |
|
Balanced calcium intake (1.2 g/day) / calcium supplement |
|
Hypocitraturia (< 1.5 mmol/day) |
|
Low animal protein intake (0.8 – 1 g/Kg/day) |
|
High intake of fruits and vegetables / potassium, citrate and magnesium supplementation |
|
Low urine pH / hyperuricosuria (> 4.5 and 4.8 mmol/day for women and men respectively) |
|
High intake of fruits and vegetables / potassium, citrate and magnesium supplementation |
|
Low dietary purine intake |
Reviewer 2 Report
Dietary manipulation is at the fore-front of measures to prevent recurrence of kidney stones. This review purports to analyse the effects of dietary factors individually on stone incidence & recurrence, & discusses the principles underlying current recommended dietary guidelines. The lack of data for vegans diets was highlighted. It is not a systematic review presented according to PRISMA guidelines, but is closer to an editorial or commentary. It is generally clearly written with compact, useful Tables, & a comprehensive reference list featuring mainly recent articles.
General comments
- Several recent reviews have covered this topic with more detailed coverage of the studies to investigate the efficacy of intervention, notably: d'Alessandro et al, Nutrients2019 (Ref 107 in this manuscript [M/S], Prezioso et al (M/S ref 23). Heidelberg (M/S Ref 61), Gambaro (M/S Ref 27); also: Goldfarb DS Urolithiasis 2019;47: 107-13 & American Urol Assn : medical management of stones, 2019 update. There have been no significant advances in management since these publications.
- The Authors rightly stress the value of metabolic investigation, focusing on 24h urine analyses, to define the phenotype of stone formers (sfs) & enable tailored changes to diet (L 73-87). However, world-wide only a small proportion of sfs are investigated. Hence there is a need for a diet which can be applied 'blindly' to the remainder, that is effective, and safe (does not provoke stone formation, bone demineralisation or poor growth [children]). This requires more emphasis.
- As background to this empiric approach, it would help to point out that heritability accounts for around 50% of the risk for idiopathic stones, that in most cases this is polygenic with variations in numerous genes potentially interacing in different individuals. The influence of diet on urine composition will vary with genetic make-up. Except when monogenic disorders are suspected, genetic testing is not an option to guide dietary manipulation. Genetic predisposition was not mentioned.
- To be useful to those managing sfs, more precise information should be relayed from the published studies, in order to assess the merits/risks of intervention. Who was studied? nationality? how many? sex? what was the actual reduction in stone incidence ? what was the quantitative effect of a dietary manouvre on urine biochemistry? How much does urine calcium increase when dietary Ca is raised eg from 400mg to 1200mg /day in sfs with normocalciuria, or those with hypercalciuria, or those taking Vit D? There were some quantitative data for oxalate & sodium. Among papers needing more detailed cover were Ref 36 (L99-102), Ref 65 (L189-192), Ref46 (L236), Ref77 (L268), Ref 90 (L270), Ref 59 (L312-314)
- L237-242 Although Mg is generally listed with inhibitors of Ca Oxalate stone formation because of in vitro testing, its importance as a protective agent in vivo is disputed. Only one study has demonstrated an association between urine Mg & stone formation [in male but not female adults] (Taylor EN et al J Am Soc Nephrol 2004; 15: 3225-32)
Omissions
1. No information about how publications were selected for review, or reasons for exclusion.
2. No discussion of whether/how the dietary recommendations should be modified for children, adolescents, the elderly & in pregnancy
Some specific points
L94-98-Confusing to introduce hyperparathyroidism & osteoporosis
From L124; ? Need to expand a little: ? add that intestinal oxalate absorption is variable & normally low (around 10-15%). In individuals without GIT pathology, absorption is increased when intestinal ionised Ca is reduced (phytate, low dietary Ca; refer to Section 2.1). Absorption is also increased with formation of Ca Oxalate stones in patients with bowel disorders-enteric hyperoxaluria-but such secondary disturbances are not the topic of this review
L140 What organisms are in the VSL-3 probiotic? What is the foundation for using it? Is there any evidence that idiopathic sfs have an abnormal gut microflora?
Section 2.7 (starts L256). Beverages This para should be moved to be the first dietary modification to be discussed. It is by far the most important
L273-2 litres of lemon juice per day?-surely not fresh, undiluted, juice?
L304-'EPIC'-in full when first mentioned. Who was included inthe EPIC-Oxford study? How many subjects? age? sex?
Author Response
General comments:
1. Several recent reviews have covered this topic with more detailed coverage of the studies to investigate the efficacy of intervention, notably: d'Alessandro et al, Nutrients2019 (Ref 107 in this manuscript [M/S], Prezioso et al (M/S ref 23). Heidelberg (M/S Ref 61), Gambaro (M/S Ref 27); also: Goldfarb DS Urolithiasis 2019;47: 107-13 & American Urol Assn : medical management of stones, 2019 update. There have been no significant advances in management since these publications.
Answer: Although there were no significant changes in dietary management of kidney stones, we focused this review on the available literature of the effect of vegetarian / vegan diet on the risk of stones. To better explain their effect on kidney stones, we decided to include a brief analysis of the main dietary factors capable of influencing kidney stone recurrence.
2. The Authors rightly stress the value of metabolic investigation, focusing on 24h urine analyses, to define the phenotype of stone formers (sfs) & enable tailored changes to diet (L 73-87). However, world-wide only a small proportion of sfs are investigated. Hence there is a need for a diet which can be applied 'blindly' to the remainder, that is effective, and safe (does not provoke stone formation, bone demineralisation or poor growth [children]). This requires more emphasis.
Answer: We modified the revised manuscript as suggested, emphasizing the need of general dietary advice applicable to all stone formers, avoiding bone demineralization or stunted growth in children (L98-100 “Anyway, since metabolic evaluation is not routinely performed in all stone formers, general dietary advice capable of reducing the risk of nephrolithiasis avoiding bone demineralization or stunted growth in children are fundamental for kidney stones patients.”)
3. As background to this empiric approach, it would help to point out that heritability accounts for around 50% of the risk for idiopathic stones, that in most cases this is polygenic with variations in numerous genes potentially interacing in different individuals. The influence of diet on urine composition will vary with genetic make-up. Except when monogenic disorders are suspected, genetic testing is not an option to guide dietary manipulation. Genetic predisposition was not mentioned
Answer: We followed the suggestion of the reviewer to add information about genetic predisposition (introduction section). We also focalized the attention on the lack of genetic work-up for polygenic predisposition, giving greater importance to tailored dietary prescription based on 24h urine analysis, since the potential variable effect of diet in this contest (L48-50 “Furthermore, genetic predisposition should also be considered. More than 30 genetic variants with Mendelian inheritance are known for causing kidney stones and polygenic involvement in idiopathic stone formers is even more frequent” and L95-97 “In addition, genetic work-up is available only for selected, monogenic forms of nephrolithiasis, thus personalized dietary indications has even greater importance in case of polygenic predisposition, since the potential variable effect of diet in this context.”).
4. To be useful to those managing sfs, more precise information should be relayed from the published studies, in order to assess the merits/risks of intervention. Who was studied? nationality? how many? sex? what was the actual reduction in stone incidence? what was the quantitative effect of a dietary manouvre on urine biochemistry? How much does urine calcium increase when dietary Ca is raised eg from 400mg to 1200mg /day in sfs with normocalciuria, or those with hypercalciuria, or those taking Vit D? There were some quantitative data for oxalate & sodium. Among papers needing more detailed cover were Ref 36 (L99-102), Ref 65 (L189-192), Ref46 (L236), Ref77 (L268), Ref 90 (L270), Ref 59 (L312-314)
Answer: We added more precise information on the effect of calcium intake on urinary excretion of oxalate and for the suggested references (L119-126 “In detail, a study on three large cohorts (the Health Professionals Follow-Up Study, Nurses’ Health Study I and Nurses’ Health Study II) on a total of 194,095 participants and a median follow-up of more than 8 years, showed higher risk for incident kidney stones for the consumption of one or more servings of sugar-sweetened non-cola per day compared to less than one per week (+33%, 95% confidence interval [CI] = 1% to 74%), followed by sugar-sweetened cola (+23%) and punch (+18%). On the contrary, an analysis of the same cohorts suggested significantly lower risk of kidney stones for people in the highest quintile of caffeinated coffee consumption (> 500 mg/day of caffeine, −26%, −29% and −31% depending on the cohort)”, L150-154 “Significant insights on the role of calcium in stone disease were obtained in 1993, when an analysis of 45,619, 40-75 years old male individuals without stone disease (the Health Professionals Follow-up Study cohort) showed that lower calcium intakes were associated with higher risk of kidney stone events by more than 50% compared with higher dietary intakes (797 ± 280 vs 851 ± 307 mg of dietary calcium respectively)”, L154-161 “This was confirmed in a randomized trial comparing a normal calcium (1200 mg/day), low salt and low animal protein diet to a low calcium diet (400 mg/day) in a group of 120 men with recurrent calcium oxalate stones and hypercalciuria. In this study, the normal calcium diet group demonstrated a significant reduction in the risk of stone recurrence of approximately 50% after five years compared to the low calcium diet group (relative risk for recurrence 0.49, 95% CI 0.24 to 0.98). In addition, urinary oxalate excretion was higher in patients on low dietary calcium intake (60 μmol/day increase) and lower in patients under normal calcium, low salt and low animal protein diet (80 μmol/day decrease)”, L258-251 “Giannini et al. demonstrated that in 18 hypercalciuric stone formers a protein intake of 0.8 g/kg/day significantly improved several urinary lithogenic risk factors, i.e. decreased urinary calcium (from 9.35 ± 0.3 to 6.45 ± 0.3 mmol/day) and uric acid excretion (from 3.1 ± 0.1 to 2.5 ± 0.1 mmol/day) and increased urinary citrate excretion (from 3.42 ± 0.3 to 5.34 ± 0.9 mmol/day)”, L287-297 “Borghi et al. in 2004 analyzed the effect of dietary alkali load on urinary composition in both healthy individuals and stone formers. They proved that withdrawing fruits and vegetables from diet for 14 days in 12 normal subjects significantly reduce urinary excretion of magnesium (−26%), citrate (−44%), potassium (−62%) and increases urinary calcium (+49%) and ammonium (+12%). Urine supersaturation for calcium oxalate and phosphate increases as well (from 6.33 to 8.24 and from 0.68 to 1.58 respectively). On the contrary, a diet rich in fruits and vegetables in hypocitraturic stone formers increases urinary pH (from 5.84 to 6.19), excretion of potassium (+68%), citrate (+58%) and magnesium (+23%), and reduces ammonium (−18%) whereas urinary excretion of calcium and oxalate remained unchanged. These urinary modifications led to reduced supersaturation for uric acid and calcium oxalate (from 10.17 to 4.96 and from 2.78 to 1.12 respectively)”, L351-357 “These observations were confirmed by studies investigating the effect of the DASH-style and the Mediterranean diets on renal stones. Although allowing moderate meat consumption, the high content of vegetables and low in animal proteins of subjects in the highest quintile of the DASH-diet score (composed of 8 items: intake of fruits, vegetables, nuts and legumes, low-fat dairy products, whole grains, intake of sodium, sweetened beverages and red meats) was demonstrated to be associated with a lower relative risks for kidney stones (0.55, 0.58 and 0.60 in men, older and younger women respectively) compared with regular American diet”).
5. L237-242 Although Mg is generally listed with inhibitors of Ca Oxalate stone formation because of in vitro testing, its importance as a protective agent in vivo is disputed. Only one study has demonstrated an association between urine Mg & stone formation [in male but not female adults] (Taylor EN et al J Am Soc Nephrol 2004; 15: 3225-32)
Answer: We clarified the role of magnesium on kidney stones risk, as suggested (L304-307 “Despite the strong rationale for considering magnesium a kidney stone inhibitor, available literature is controversial. Early observational evidence showed no significant association between magnesium and the risk of stones, whereas Taylor et al in 2004 demonstrated an inverse correlation between the risk of incident kidney stones and dietary magnesium intake”).
Omissions
1. No information about how publications were selected for review, or reasons for exclusion.
Answer: We modified the revised manuscript as suggested, adding a methods section (L70-76 “In this review, we included all the available literature regarding the association between vegetarian and vegan diets and kidney stone disease. We selected articles, with unrestricted search period, on several databases including PubMed, Google Scholar, the Cochrane library and Web of Science. To identify articles of interest we used the following search terms: “kidney stones” OR “nephrolithiasis” OR “urolithiasis” OR “kidney calculi” AND “diet”, “dietary indications”, “dietary advice”, “intake”, “vegetarian”, “vegan”. We included in this review only articles written in English language and with available full-text. No restriction was made based on the type of article.”).
2. No discussion of whether/how the dietary recommendations should be modified for children, adolescents, the elderly & in pregnancy
Answer: We agree with the reviewer that recommendations for those groups would be of great interest; however, we believe that to date, there is not enough evidence for optimizing dietary recommendations for those groups
Some specific points
L94-98-Confusing to introduce hyperparathyroidism & osteoporosis
Answer: We removed the confusing sentence.
From L124; ? Need to expand a little: ? add that intestinal oxalate absorption is variable & normally low (around 10-15%). In individuals without GIT pathology, absorption is increased when intestinal ionised Ca is reduced (phytate, low dietary Ca; refer to Section 2.1). Absorption is also increased with formation of Ca Oxalate stones in patients with bowel disorders-enteric hyperoxaluria-but such secondary disturbances are not the topic of this review
Answer: we added the suggested sentence (L186-189 “Normally, intestinal oxalate absorption is low and highly variable (around 10-15%). In individuals without malabsorption syndrome, bowel oxalate uptake may increase only when intestinal ionized Ca is reduced, often due to both high dietary consumption of phytate (calcium binding) or low calcium diet”).
L140 What organisms are in the VSL-3 probiotic? What is the foundation for using it? Is there any evidence that idiopathic sfs have an abnormal gut microflora?
Answer: We specified the composition of VSL-3 (mixture of 8 strains of lactic acid–producing bacteria), the rationale (although controversial), is the modulation of intestinal oxalate absorption. We further cited evidence of different gut microbiome composition in stone formers vs non-stone formers (L201-203 “However, although differences in gut microbiome composition between stone formers vs non-stone formers were recently found [ref49], there is still lack of solid scientific evidence for the effectiveness of probiotics in reducing the urinary excretion of oxalate.”).
Section 2.7 (starts L256). Beverages This para should be moved to be the first dietary modification to be discussed. It is by far the most important
Answer: We modified the revised manuscript as suggested.
L273-2 litres of lemon juice per day?-surely not fresh, undiluted, juice?
Answer: A solution composed of 120 mL of concentrated lemon juice and 2 L of water. Added to the manuscript (L131)
L304-'EPIC'-in full when first mentioned. Who was included inthe EPIC-Oxford study? How many subjects? age? sex?
Answer: We modified the revised manuscript as suggested, including the requested information (L344-347 “This study was conducted on 51,336 participants (23% were male) aged ≥ 20 years. A total of 303 episodes of incident kidney stones were recorded (716,105 person-years of follow-up). It turned out that, compared to high meat intake diet, vegetarian and low meat intake diets had the lowest hazard ratio: 0.69 (95% CI 0.48, 0.98) and 0.52 (95% CI 0.35, 0.80) respectively”).
Reviewer 3 Report
The authors have provided a comprehensive review of dietary factors associated with kidney stone disease. The information contained will be useful to dieticians, nutritionists and clinicians who treat stone disease. Their conclusion that stone formers would benefit from consuming a balanced vegetarian diet with dairy products seems warranted from the extensive evidence they have presented.
Author Response
I like to thank you for your diligent work on our paper.
Round 2
Reviewer 2 Report
The Authors have answered/addressed all of my queries and, in my view, have enhanced the value of the Manuscript to others. I note that they have added a new Table (Table 2) which shows helpful guidelines
Minor comments/suggestions
L97 -Delete 'in addition'. Start 'Genetic--'
L98-99 Reword after 'indications' for easier reading; eg to 'For those with polygenic disposition, diet may have a variable effect and personalised dietary advice is desirable. However, since metabolic---etc'
L121 Change 'on' to 'of'
L133 Please check recipe. Should it be 120ml made up to 2 litres?; change 'was able to' to 'reduced'
L139 sp. 'havinge'-delete 'e'
LI55 'without stone disease'-was this at recruitment?
LI66 ? change 'at the intestinal level' to 'in the intestine ---'
L188 delete 'both' & L189 change 'or' to 'and/or'
L247 'We previously---'-wrong Ref; not 29-should it be 30 (Ferrari)?
L253 after 0.8g/kg/day ? add 'and 955mg of calcium for 15 days'
L254 urinary oxalate also decreased; add the values after urinary calcium
L291 although the study was by Borghi's Group, the publication was by Meschi et al (ref 46)
L294 oxalate decreased too-should add
L294/295 observations from one study, so change to past tense: 'reduce' to 'reduced', 'increases' to 'increased' etc
L298 citrate increased by +68%, not +58%
L386-391appears to report one large study (Taylor et al; Ref 60). It needs re-writing to make sense. eg 'In one large prospective study of 241,766 male & female health workers in the USA a score for the DASH-style diet was devised based on 8 components (intake of fruit-etc, etc). This was used to assess the diet over 14-18 years. Comparing individuals in the highest & lowest DASH score quintiles, those with the highest scores had significantly lower risk for developing kidney stones. Multivariate relative risks for stones were--etc, etc'
Text numbering issues
L355 sub-heading 3.1 should be 4.1
L414 sub-heading 4 (Conclusions) should be 5
Reference numbers. Many are wrong from around Ref 29 because the section on fluid intake was moved without re-numbering the references
Author Response
REVIEWER 2 – REPORT 2
Minor comments/suggestions
L97 -Delete 'in addition'. Start 'Genetic--' Answer: We modified the revised manuscript as suggested. (L95)
L98-99 Reword after 'indications' for easier reading; eg to 'For those with polygenic disposition, diet may have a variable effect and personalised dietary advice is desirable. However, since metabolic---etc' Answer: we modified the sentence as follow “thus for those with polygenic disposition, diet may have a variable effect and personalized dietary advice is desirable. However, since metabolic evaluation is not routinely performed in all stone formers, general dietary advice capable of reducing the risk of nephrolithiasis avoiding bone demineralization or stunted growth in children are fundamental for kidney stones patients” (L96-97)
L121 Change 'on' to 'of' Answer: we modified the revised manuscript as suggested “In detail, a study of three large cohorts” (L118)
L133 Please check recipe. Should it be 120ml made up to 2 litres?; change 'was able to' to 'reduced'
Answer: sentence changed to “120 mL of concentrated lemon juice, diluted to a 2 L solution reduced….” (L130)
L139 sp. 'havinge'-delete 'e' Answer: We modified the revised manuscript as suggested.
LI55 'without stone disease'-was this at recruitment? Answer: added “without stone disease at recruitment” (L150)
LI66 ? change 'at the intestinal level' to 'in the intestine ---' Answer: We modified the revised manuscript as suggested. (L161)
L188 delete 'both' & L189 change 'or' to 'and/or' Answer: We modified the revised manuscript as suggested (L183-184)
L247 'We previously---'-wrong Ref; not 29-should it be 30 (Ferraro)? Answer: we changed the wrong reference (L242)
L253 after 0.8g/kg/day add 'and 955mg of calcium for 15 days' Answer: We modified the revised manuscript as suggested. (L248)
L254 urinary oxalate also decreased; add the values after urinary calcium Answer: We changed the suggested sentence as follow: “decreased urinary calcium (from 9.35 ± 0.3 to 6.45 ± 0.3 mmol/day), oxalate (from 0.59 ± 0.09 to 0.31 ± 0.03 mmol/day) and uric acid excretion (from 3.1 ± 0.1 to 2.5 ± 0.1 mmol/day) and increased urinary citrate excretion (from 3.42 ± 0.3 to 5.34 ± 0.9 mmol/day)” (L249-251)
L291 although the study was by Borghi's Group, the publication was by Meschi et al (ref 46) Answer: changed in “Meschi et al” (L287)
L294 oxalate decreased too-should add Answer: “oxalate (-31%)” added (L291)
L294/295 observations from one study, so change to past tense: 'reduce' to 'reduced', 'increases' to 'increased' etc Answer: We modified the revised manuscript as suggested (L290-297)
L298 citrate increased by +68%, not +58% Answer: We modified the revised manuscript as suggested (L294)
L386-391appears to report one large study (Taylor et al; Ref 60). It needs re-writing to make sense. eg 'In one large prospective study of 241,766 male & female health workers in the USA a score for the DASH-style diet was devised based on 8 components (intake of fruit-etc, etc). This was used to assess the diet over 14-18 years. Comparing individuals in the highest & lowest DASH score quintiles, those with the highest scores had significantly lower risk for developing kidney stones. Multivariate relative risks for stones were--etc, etc' Answer: We changed the sentence in “In one large prospective study of 241,766 male & female health workers in the USA a score for the DASH-style diet was devised based on 8 components (intake of fruits, vegetables, nuts and legumes, low-fat dairy products, whole grains, intake of sodium, sweetened beverages and red meats). This was used to assess the diet over 14-18 years. Comparing individuals in the highest & lowest DASH score quintiles, those with the highest scores had significantly lower risk for developing kidney stones. Multivariate relative risks for stones were 0.55, 0.58 and 0.60 in men, older and younger women respectively” (L352-358)
Text numbering issues
L355 sub-heading 3.1 should be 4.1 Answer: We modified the revised manuscript as suggested
L414 sub-heading 4 (Conclusions) should be 5 Answer: We modified the revised manuscript as suggested
Reference numbers. Many are wrong from around Ref 29 because the section on fluid intake was moved without re-numbering the references Answer: references updated as suggested